

# Methane, carbon dioxide, hydrogen sulfide, and isotopic ratios of methane observations from the Permian Basin tower network

Vanessa C. Monteiro[1], Natasha L. Miles[1], Scott J. Richardson[1], Zachary Barkley[1], Bernd J. Haupt[2], David Lyon[3], Benjamin Hmiel[3], Kenneth J. Davis[1]

[1]Department of Meteorology and Atmospheric Science, The Pennsylvania State University, PA
[2]Earth and Environmental Systems Institute, The Pennsylvania State University, PA
[3]Environmental Defense Fund, Austin, TX, USA

*Correspondence to*: Natasha Miles (nmiles@psu.edu)

**Abstract.** We describe the instrumentation, calibration, and uncertainty of the network of ground-based, in situ, cavity ring down spectroscopy (CRDS) greenhouse gas (GHG) measurements deployed in the Permian basin. The primary goal of the network is to be used in conjunction with atmospheric transport modeling to determine methane emissions of the Delaware sub-basin of the Permian Basin oil and natural gas extraction area in Texas and New Mexico. Four of the measurements are based on tall communications towers, while one is on a building on a mountain ridge, with the recent addition of a small tower at that site. Although methane ($CH_4$) is the primary specie of interest, carbon dioxide ($CO_2$), hydrogen sulfide ($H_2S$), and the isotopic ratio of methane ($\delta^{13}CH_4$) are also reported for a subset of the sites. Measurements were reported following WMO X2004A scale for $CH_4$, and the WMO X2019 scale for $CO_2$. CRDS instruments were calibrated for $CH_4$ and $CO_2$ in laboratory prior deployment. For $H_2S$, data was offset-corrected using the minimum 40-min running mean value of the day, and for $\delta^{13}CH_4$, calibrations were based on laboratory data. We describe the characteristics of the data set with a set of illustrative analyses. Methane and carbon dioxide showed strong seasonality, with a well-defined diurnal cycle during the summer, which was opposed to the winter, when a diurnal cycle was absent. $CH_4$ enhancements to the background, during the winter, are up to twice the summer values, which is attributed to the changes in boundary layer depth and wind speed. The largest $CH_4$ enhancements occurred when winds blow from the center of the Delaware sub-basin, where most of the methane emissions come from. The magnitude of enhancements of $CO_2$ did not present seasonality. $H_2S$ enhancements indicated a potential source northeast of the tower where the inlet is installed. Isotopic ratios of methane indicated that oil and natural gas extraction is the source of local methane in the region. The hourly-averaged data, starting on 1 March 2020 and described in this paper, are archived at The Pennsylvania State University Data Commons at https://doi.org/10.26208/98y5-t941 (Monteiro et al., 2021).

## 1 Introduction

Emissions of methane ($CH_4$), such as from oil and natural gas producing regions, are an environmental concern since $CH_4$ is a greenhouse gas with a global warming potential 28-36 times larger than that of carbon dioxide ($CO_2$) over a 100-year



period, and 80 times larger than CO2 over a 20-year period (IPCC, 2021). This large difference in radiative forcing is a result of the relatively short atmospheric lifetime of $CH_4$ (~ 10 years) compared to $CO_2$ (lifetime of ~ hundreds or thousands of years, IPCC, 2021). As a result of its strong short-term impact, reductions in $CH_4$ emissions are an efficient way to quickly reduce radiative forcing. According to Ocko et al. (2021), dramatic methane mitigation measures now could decrease in 30

% the global-mean rate of near-term global temperature increase.

About 60 % of the global $CH_4$ budget arises from anthropogenic emissions (Saunois et al., 2020). Some of these emissions are fairly well-known since they are large point sources amenable to direct emissions monitoring (e.g., coal mines) (Kirchgessner et al., 2000). Some are relatively diffuse, large-area, low-intensity sources such as agricultural activities

(Carlson et al., 2017; Moraes et al., 2013). A large component of anthropogenic emissions, however, comes from relatively compact, high-intensity, regional sources such as oil and gas (O&G) production basins (Alvarez et al., 2018; Maasakkers et al., 2019; Pandey et al., 2019; Schwietzke et al., 2016). O&G producing regions include numerous point sources covering a wide range of expected emissions (e.g., well pads and processing plants) and more diffuse sources as gathering pipelines, sometimes intermingled with other methane sources such as livestock. Thus, O&G emissions are large in magnitude, and

often highly uncertain. In addition, O&G production in the United States has increased dramatically since around 2005, driven primarily by hydraulic fracturing and horizontal drilling (Alvarez et al., 2012). This expansion of O&G production has prompted an increasing interest in monitoring of methane emissions from these basins for regulations and commercial incentives for operators to prove low emissions.

Atmospheric $CH_4$ measurements can reduce uncertainty in $CH_4$ emissions by providing "top-down" assessments of emissions. Top-down estimates are based on empirical data and atmospheric scientific methods, opposed to "bottom-up", which uses an inventory approach and extrapolate regional emissions from smaller spatial scale measurement data as a component. The top-down emissions estimates can be compared with and used to improve more traditional accounting-based, or inventory methods (e.g., Maasakkers et al., 2016; U.S. EPA, 2019). Emissions of $CH_4$ in the U.S. derived from

atmospheric data have differed significantly from inventory assessments (Alvarez et al., 2018; Turner et al., 2015; Barkley et al., 2019b; Barkley et al., 2021; Zhang et al., 2020), showing the importance of such independent data. Many past atmospheric studies of $CH_4$ have used aircraft data to quantify emissions from O&G production basins (Baray et al., 2018; Barkley et al., 2017; 2019a; 2019b; Karion et al., 2015; Schwietzke et al., 2017) and cities (Cambaliza et al., 2014; Conley et al., 2016; Heimburger et al., 2017; Plant et al., 2019). Automobile-based measurements (e.g., Caulton et al., 2019; Omara et

al., 2016; Robertson et al., 2020) have also been used to great advantage to characterize emissions from individual O&G production sites. Aircraft and automobile measurements, which are spatially rich in information content, are typically short-term in nature. Even extended airborne campaigns (e.g., Heimburger et al., 2017; Barkley et al., 2021) have limited availability to date to capture temporal trends in basin- or city-scale emissions.



Observations of atmospheric methane are sparse, with limited in situ sites located mostly in North America and Europe (e.g., Andrews et al., 2014; Karion et al., 2015). Satellite-based measurements such as the Greenhouse Gas Observing Satellite (GOSAT) and the TROPOspheric Monitoring Instrument (TROPOMI) data provide global, remote sensing of column atmospheric methane ($XCH_4$; e.g., Qu et al., 2021), and will strengthen our understanding of the global methane budget, but quantification of the level of bias and uncertainty in methane emissions estimates as a function of conditions known to affect

satellite retrievals is critical. Tower- and building-based in situ networks measuring $CH_4$ dry mole fractions within the boundary layer have been used to quantify urban emissions (e.g., Lamb et al., 2016; McKain et al., 2015; Yadav et al., 2019; Sargent et al., 2021).  Lin et al. (2021) quantified $CH_4$ emissions from oil and gas facilities in the Uinta Basin using in situ observations. Here, we present observations from a tower-based atmospheric monitoring network designed to track $CH_4$ emissions from the Delaware sub-basin of the Permian Basin.


The Permian Basin is the largest-producing region of oil in the United States, and the second largest-producing region of natural gas, accounting for 40% of the U.S. oil production and 15 % of the U.S. natural gas production (Enverus Drillinginfo, (2021)). The Permian Basin is also a large emitting U.S. oil and natural gas producing region according to satellite observations (Zhang et al., 2020), with a natural gas productions normalized loss rate of 3.7 %. Emissions of methane are

associated with midstream processing (e.g., Mitchell et al., 2015), flares (e.g., Allen et al., 2016), and with low-producing marginal wells (e.g., Deighton et al., 2020).

This tower-based network was deployed in late February 2020, just prior to the COVID-19 pandemic lockdown in the United States. The data reported here stops on 9 November 2021, but the network is still in operation.  Using these tower

observations alongside aircraft measurements, satellite observations, and model estimates, Lyon et al. (2021) found a correlation between decreasing oil prices and the $CH_4$ emissions during the COVID-19 lockdown (March - April 2020), and hypothesized that under normal conditions, production exceeds the midstream capacity, resulting in more venting/flaring, and consequently higher methane emissions.

The primary purpose of this paper is to describe the high-accuracy mole fraction measurements of $CH_4$ including the network and site characteristics (Section 2), instrumentation used for data collection (Section 3), as well as the associated calibration, processing, uncertainties, and data coverage (Section 3). We also describe opportunistic measurements of $CO_2$, hydrogen sulfide ($H_2S$), and methane isotope ratio ($\delta^{13}CH_4$) which are of interest but are not the focus of this network. Section 4 presents summary data for all gases ($CH_4$, $CO_2$, $H_2S$, and $\delta^{13}CH_4$) to date, and an example analysis of each gas.

The example analyses include diurnal cycles and enhancements for $CH_4$, $CO_2$, and $H_2S$, and the determination of source isotopic signature using $\delta^{13}CH_4$ and $CH_4$ measurements. We have included brief methods for these analyses in each subsection, rather than a separate section for methods.

## 2 Network and sites characterization

The Permian in situ tower observation network includes five monitoring stations in the Delaware sub-basin (Fig. 1a, Table
1): Carlsbad Caverns National Park (CARL), Maljamar (MALJ), Hobbs (HOBB), Notrees (NOTR), and Fort Stockton
(FORT). The instruments installed at the stations measure methane concentrations continuously, beginning 1 March 2020.
The towers encompass an area of approximately 160 km x 220 km.  Of the monitoring stations, four are communications
towers with gas inlets installed at 91 – 134 m height above ground level (AGL).  Due to the lack of availability of a tower,
the instrumentation at the Carlsbad location was initially deployed on a rooftop (4 m AGL). On 13 May 2021, the
instrumentation was moved to a 9 m tower approximately 250 m to the east of the original location. The Carlsbad site is at
1349 m above sea level on a mountain ridge, significantly higher than the surroundings' elevation (e.g., the elevation in
White City, 6 km to the east of Carlsbad tower, is 1112 m ASL). The location is within Carlsbad Caverns National Park,
hence is buffered from oil and gas infrastructure immediately adjacent to the measurement site.

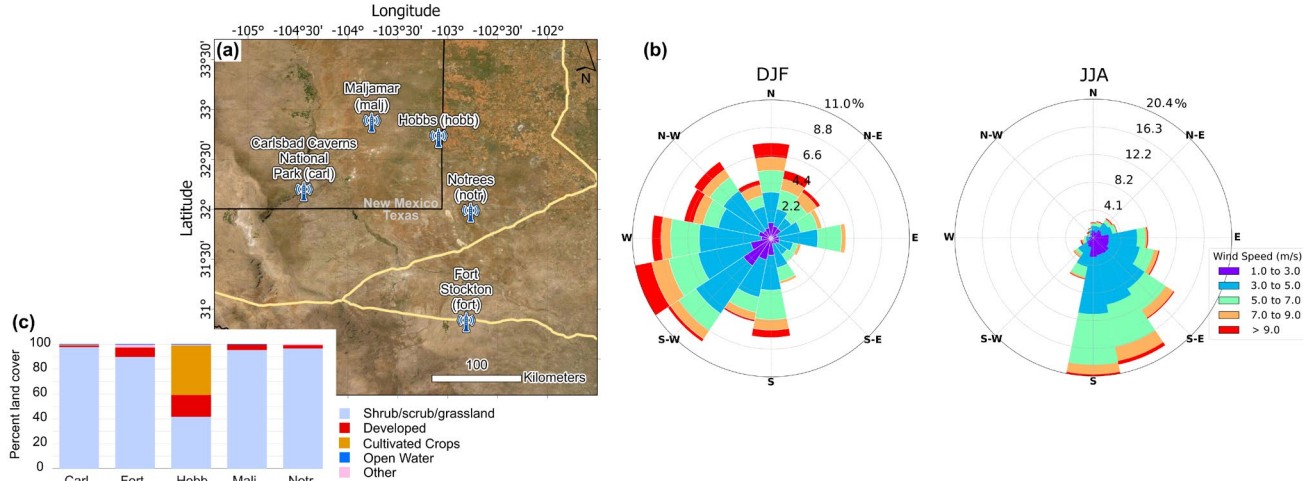


**Figure 1: Location of the Permian network towers and land cover characteristics. (a) Five towers located in New Mexico and Texas as part of the Permian towers network. The towers continuously measure $CH_4$. Additionally, CARL, FORT, MALJ measure $CO_2$, HOBB measures $H_2S$, and MALJ and NOTR measure $\delta^{13}CH_4$. (b) Wind rose showing the prevailing wind direction during winter months, i.e., December, January, February (DJF), and summer months, i.e., June, July, August (JJA) for the years of 2020
and 2021. The percent scale (radial axis) shows the frequency of the wind blowing from a specific direction. Weather data were obtained from Lea County Regional Airport in Hobbs, NM (retrieved from Iowa Environmental Mesonet (IEM, 2021)). (c) Percent land cover from 2016 National Land Cover Database (MRLC, 2019) within 10 km radius of each tower. Most of the area surrounding the towers is covered by shrub, scrub, and grassland. Some land cover types were grouped for simplicity, e.g., "developed" corresponds to all developed categories (high density, medium density, low density, and open space), and "other"
corresponds to evergreen forest, barren land, woody wetlands, and emergent herbaceous wetlands. Credits for basemap: Esri, DigitalGlobe, GeoEye, Earthstar Geographics, CNES/Airbus DS, USDA, USGS, AeroGRID, IGN, and the GIS User Community.**





**Table 1: Locations, inlet heights, species measured, and installation dates of in-situ tower-based measurements in the Permian Basin.**

| Site | Latitude | Longitude | Elevation (m ASL) | Inlet height (m AGL) | Species measured | Install date |
|---|---|---|---|---|---|---|
| Carlsbad Caverns National Park (CARL) | 32.1783 N | 104.4406 W | 1349 | 4, 9 | $CH_4$, $CO_2$ | 29 Feb 2020 |
| Fort Stockton (FORT) | 30.8666 N | 102.8150 W | 987 | 128 | $CH_4$, $CO_2$, $\delta^{13}CH_4$ | 29 Feb 2020 |
| Hobbs (HOBB) | 32.7135 N | 103.0913 W | 1103 | Inside (0.5), 2, 91 | $CH_4$, $CO_2$, $H_2S$ | 27 Feb 2020 |
| Maljamar (MALJ) | 32.8671 N | 103.7608 W | 1310 | 134 | $CH_4$, $CO_2$, $\delta^{13}CH_4$ | 27 Feb 2020 |
| Notrees (NOTR) | 31.9657 N | 102.7699 W | 1015 | 91 | $CH_4$, $CO_2$, $\delta^{13}CH_4$ | 28 Feb 2020 |

The prevailing wind direction in the region varies seasonally (Fig. 1b). During the winter (i.e., December, January, February (DJF)), the wind most often comes from SW - W directions, while during the summer (i.e., June, July, August (JJA)), the prevailing direction is from the South. This behavior makes the measurements obtained from Carlsbad (during winter) and

Fort Stockton (during summer) the most likely background sites since most of the time they are not directly impacted by $CH_4$ emissions originating from the oil and gas fields located within the Delaware sub-basin to the east and north of these towers, respectively.

New Mexico and Texas are within a region of extensive production of oil and natural gas, and the landscape surrounding the

towers are mostly shrub/scrub, and grassland (Fig. 1c). Hobbs is the only site with significant agricultural and urban landcover. Within a 10 km radius of the Hobbs tower, the landscape is ~ 40 % shrub, scrub, and grassland, ~ 40 % cultivated crops (to the east of the tower), and ~ 20 % urban area (to the west of the tower). These relatively simple surroundings in terms of methane emissions simplifies the task of isolating emissions from the basin's O&G infrastructure, which accounts for > 90 % of methane emissions in the Permian Basin (Maasakkers et al., 2016).



## 3 Instrumentation, calibration measurements, uncertainty

### 3.1 Instrumentation and calibration

Mole fraction measurements were made with wavelength-scanned cavity ring down spectroscopic (CRDS) instruments (Picarro, Inc., models G2301, G2401, G2204, and G2132-i). The primary species of interest for this network was $CH_4$. Most of the instruments also reported $CO_2$, one reported $H_2S$ and, at various locations and time periods, $\delta^{13}CH_4$ was reported. Instrument failures necessitated multiple exchanges of instrumentation for repairs.

The in situ sampling method was similar to the procedures described in Richardson et al. (2017) and the schematic for the systems as deployed in the field is shown in Fig. 2. Collocated at the top sampling level of each tower were two 1/4 in (0.64 cm) OD Synflex 1300 (Eaton Corp.) tubes with rain shields to prevent liquid water from entering the sampling line. Air was drawn down from the inlet on the tower, through the Nafion dryer (MD Series, 24 in (61 cm) to 96 in (244 cm) lengths, Permapure LLC), into the CRDS instrument for analysis, and then used as the purge gas in the Nafion dryer (i.e., re-flux method). Field calibration tank gas was introduced upstream of the dryer, humidifying the calibration gas. 1/8 in (0.32 cm) OD stainless steel tubing, Air Liquide (formerly Scott Specialty Gas) regulators (part number 51-14 A-590) were used for sampling the field calibration tanks. 3-way solenoid valves (part number 091-0094-900, Parker Hannifin Corp.) were used to switch between sample and field calibration gas. Correction factors were applied to adjust the $CH_4$ and $CO_2$ values for the effects of the remaining water vapor (Rella et al., 2013).

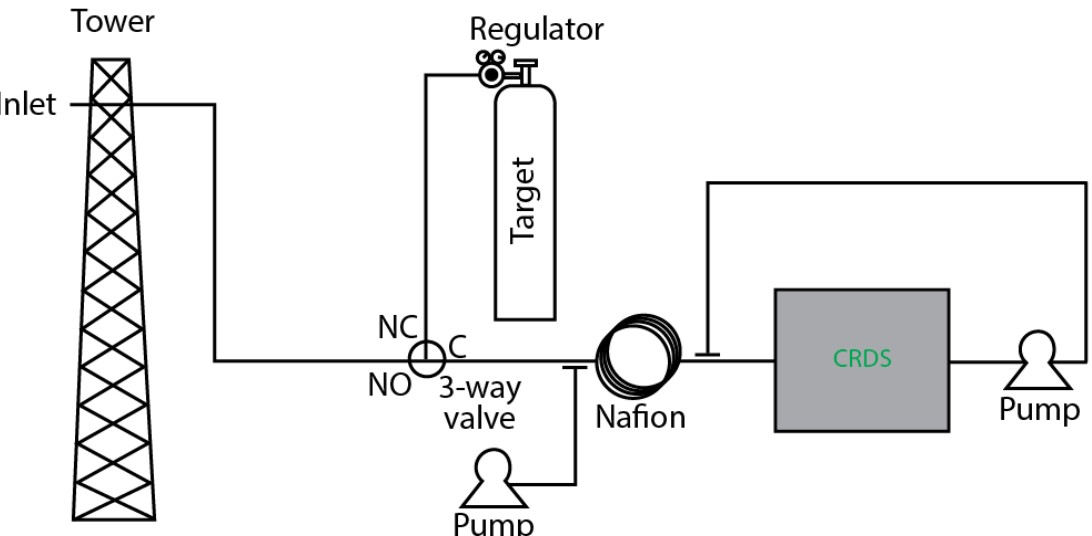

**Figure 2: Schematic of the systems deployed in the field.**



The measurements are reported on the World Meteorological Organization (WMO) X2004A scale for $CH_4$, and the WMO
X2019 scale for $CO_2$. $H_2S$ and $\delta^{13}CH_4$ are reported, but with limited calibration. The CRDS instruments were calibrated for
$CH_4$ and $CO_2$ in the laboratory prior to deployment using four NOAA (National Oceanic and Atmospheric Administration)
tertiary standards, ranging between 1790 and 2350 ppb $CH_4$, and 360 and 450 ppm $CO_2$. Field calibration tanks (Table 2)
were sampled nominally every 23 hours for 6 min for G2301, G2401 and G2204 instruments, and every 6 hours for 20 min
for G2132-i instruments, but adjustments were occasionally made to limit field calibration gas usage. After each transition
between calibration gas and atmospheric sample, 4 min of data are ignored. An offset correction was applied daily. One
disadvantage of this procedure is the potential introduction of tank drift to the data, but tank drift for $CH_4$ has not been
observed (Andrews et al., 2014). Allan deviations were < 0.2 ppb $CH_4$ and 0.02 ppm $CO_2$ for 2 min samples for the G2301
and G2401 instruments (Yver Kwok et al., 2015), and 0.1 ppb $CH_4$ and 0.1 ppm $CO_2$ for ~ one hour sample for the G2132-i
instruments (Miles et al., 2018) and thus the noise for these instruments for the calibration cycles is insignificant. Ideally,
more than one tank would be sampled at each site, but this was not practical for this network. Data from the first four
minutes after a transition between gases was discarded. Flow rates for G2301, G2401 and G2204 instruments were about 240
cc/min whereas the flow rates for the G2132-i instruments were about 30 cc/min. Residence time from the top of the tower to
measurement was 6 to 9 min for the G2301, G2401, and G2204 instruments and 45 to 70 min for the G2132-i. For the
Carlsbad site sampling from a rooftop, then a 9 m tower, residence times were < 1 min. The data were adjusted to report the
sampling time, rather than the measurement time.

**Table 2: Table of calibration cylinders used at the Permian Basin Tower Network sites. Within each site, the cylinders are listed in order of use. *δ13CH4 values are based on field calibrations of the cylinders.**

| Location | Cylinder | Dates | $CH_4$ (ppb, x2004A) | $CO_2$ (ppm, x2019) | $\delta^{13}CH_4$ (‰)* | CO (ppb, x2014A) |
|---|---|---|---|---|---|---|
| Carlsbad | LL120758 | 29 February 2020 – 14 May 2021 | 1976.1 | 420.12 | N/A | 125.1 |
| Carlsbad | LL120763 | 14 May 2021 – 9 November 2021 | 2031.0 | 426.57 | N/A | 141.3 |
| Fort Stockton | LL120783 | 29 February 2020 – 15 May 2021 | 1977.8 | 419.79 | N/A | 126.6 |
| Fort Stockton | LL120762 | 29 February 2020 – 15 May 2021 | 1976.4 | 419.94 | N/A | N/A |
| Fort Stockton | LL55866 | 16 May 2021 – 9 November 2021 | 2351.6 | 430.40 | N/A | 304.7 |





| Hobbs | LL120792 | 27 February 2020 – 9 June 2020 | 2032.9 | 424.56 | N/A | N/A |
|---|---|---|---|---|---|---|
| Hobbs | LL120780 | 10 June 2020 – 15 September 2020 | 1974.3 | 419.90 | N/A | N/A |
| Hobbs | LL108056 | 16 May 2021 – 9 November 2020 | 2110.6 | 410.24 | N/A | N/A |
| Maljamar | LL120782 | 8 June 2020 – 9 November 2021 | 1974.0 | 419.54 | -46.5 | N/A |
| Maljamar | LL120789 | 8 June 2020 – 9 November 2021 | 2028.5 | 425.78 | -46.5 | 145.3 |
| Notrees | LL120799 | 28 February 2020 – 14 May 2021 | 2035.5 | 425.91 | N/A | 151.1 |
| Notrees | LL120795 | 15 May 2021 – 9 November 2021 | 2022.4 | 424.66 | -47.0 | N/A |


We offset-corrected the $H_2S$ data using the minimum 40-min running mean value of the day (assumed to be zero) instead of a field calibration tank because a tank was not available for an extended period of the deployment. For the period for which a field calibration was available, the mean was within 0.05 ppb of the lowest $H_2S$ throughout the day, which was small

compared to the observed signals. The standard deviation of the instrument drift based on the field calibration tank was ±0.43 ppb throughout the period June – August 2021. A calibration tank with known non-zero $H_2S$ mole fraction was not available, so we were not able to assess the calibration slope. We did not apply any correction to the $H_2S$ for water vapor.

We applied $\delta^{13}CH_4$ calibrations based on laboratory data (Miles et al., 2018) and tested with a tank characterized by the

Institute of Arctic and Alpine Research (INSTAAR) prior to deployment, but field calibration tanks with varying $\delta^{13}CH_4$ values were not available for this network. A daily offset-correction was applied to the $\delta^{13}CH_4$, using a field-calibrated tank. The uncertainty is estimated to be 1 ‰, compared with 0.15 ‰ uncertainty reported by Miles et al. (2018) when utilizing multiple field calibration tanks.

**3.2 Uncertainty**

The uncertainty of the reported hourly values for $CH_4$ and $CO_2$ include contributions from measurement uncertainty, extrapolation, and water vapor (Andrews et al., 2014; Verhulst et al., 2017; Karion et al., 2020). The measurement uncertainty is composed of uncertainties attributable to short-term precision, calibration baseline and scale. We assessed the effects of instrument short-term precision and drift between calibration cycles (i.e., calibration baseline) using a 31-day





running standard deviation of the daily tank residuals. Typical values vary with instrument type (e.g., 0.3 ppb $CH_4$ and 0.03
ppm $CO_2$ for G2301, 0.5 ppb $CH_4$ and 0.09 ppm $CO_2$ for G2132-i and 3.4 ppb $CH_4$ for G2204). Scale (i.e., tank assignment)
uncertainty is set to 0.3 ppb $CH_4$ and 0.03 ppm $CO_2$, following Verhulst et al. (2017). Ideally, an independent target tank is
used to independently assess the measurement uncertainty, but for this network only one tank was deployed at each site.

Because we performed a full calibration of the instruments using four NOAA-calibrated tanks prior to deployment (and upon
any factory repairs), extrapolation error is expected to be small and is not specifically reported here. Round-robin style tests
have indicated that if full calibrations are performed at least every 2 years, in addition to a daily single-point adjustment,
differences from known values are within 1 ppb $CH_4$ and 0.1 ppm $CO_2$ (Richardson et al., 2017).

We assessed the uncertainty due to water vapor based on the difference in water vapor mole fraction between the dried
sample and the humidified field calibration gas. The difference varied due to length of Nafion dryer, building temperature,
and instrument flow rate, but was typically between 0.05 and 0.7 % $H_2O$. No drying was employed at the Hobbs site for
March 2020 – May 2021, during which time the water vapor was up to 2 %. Errors in the coefficients used to determine the
water vapor correction can vary by instrument and are the largest contributor for cases with moderate or no drying. Rella et
al. (2013) showed that errors associated with the water vapor correction, even with no drying, are less than WMO
compatibility goals for $CO_2$ and $CH_4$ (0.1 ppm $CO_2$ and 2 ppb $CH_4$) if instrument specific correction factors are determined
periodically. The error associated with relying on general correction factors (as used here) is up to 0.25 ppm $CO_2$ and 2.0 ppb
$CH_4$ for 3 % water vapor. We have therefore assumed the uncertainty due to the water vapor correction to be a linear
function between these values and no error at 0 % water vapor. The uncertainty due to the water vapor correction when
drying was 0.00 – 0.06 ppm $CO_2$ and 0.0 – 0.6 ppb $CH_4$. For the period without drying at the Hobbs site, the uncertainty was
up to 0.17 ppm $CO_2$ and 1.7 ppb $CH_4$.

The initial instrument at the Notrees site did not report water vapor for 1 March – 27 July 2020 due to a laser problem. For
that period, we applied a water correction and uncertainty based on the subsequent mean and standard deviation of the water
vapor (0.53 % ± 0.19 %). The uncertainty in the water vapor value led to uncertainty in the $CO_2$ of ~ 1.1 ppm $CO_2$ and 5 ppb
$CH_4$ for this period. For 28 July – 31 December 2020, the instrument did report $H_2O$ but $CH_4$ and $CO_2$ uncertainty due to
noise continued to be higher (2.3 ppb $CH_4$ and 0.58 ppm $CO_2$) than subsequent values for this instrument (0.4 ppb $CH_4$ and
0.04 ppm $CO_2$). The intra-network $CH_4$ differences across the Delaware Basin were 135 ppb (winter) and 51 ppb (summer),
whereas the $CO_2$ differences were 0.8 ppm (winter) and 1.2 ppm (summer) (Section 4.3). Thus, the $CH_4$ uncertainty during
this period was small compared to intra-network differences, but nearly the same magnitude for $CO_2$. Consequently, we
flagged $CO_2$ for 1 March – 31 December 2020 as unsuitable for use and replaced the mole fractions with a placeholder value
(NaN). The instrument was replaced on 15 May 2021.



The results for these contributions to the uncertainty, summed in quadrature, are shown in Fig. 3. The uncertainty of most of the instruments (e.g., G2301, G2401, G2132-i) when operating normally is about 0.15 ppm $CO_2$ and 0.7 ppb $CH_4$. Note that

the Hobbs instrument does not measure $CO_2$ and that the uncertainty for $CH_4$ is larger due to the instrument type (G2204). Manufacturer precision specifications of the instrument model at this site indicate $CH_4$ precision for a 5-s sample of 2 ppb compared to the other instrument models used in this network ($CH_4$ precision for a 5-s sample of < 0.5 ppb).

**Figure 3: CH₄ and CO₂ uncertainties from March 2020 - September 2021, at the 5 sites: Carlsbad (CARL), Fort Stockton (FORT), Hobbs (HOBB), Maljamar (MALJ), and Notrees (NOTR). (a) CH₄ uncertainty. (b) CH₄ uncertainty zoomed-in (< 1 ppb). (c) CO₂ uncertainty.**


H₂S and δ¹³CH₄ are reported in this dataset but are not the focus of this research. Since a field calibration tank with non-zero

H₂S was not available, we reported the manufacturer-specified precision of 1 ppb + 0.4% H₂S as an estimate of instrument





uncertainty. The standard deviation of two months of quasi-daily field calibration data for the isotopic ratio of methane at the Maljamar site was 0.32 ‰. Because these instruments relied on a laboratory $\delta^{13}CH_4$ calibration performed in 2015, there may be additional slope errors not captured by the single field calibration tank, and we have reported the uncertainty on the isotope ratio as 1 ‰.

**4 Methane, carbon dioxide, hydrogen sulfide and isotopic ratio of methane measurements**

**4.1 Data coverage**

The data coverage of the hourly-averaged observations for each species for the Permian Basin tower network through the beginning of November 2021 is indicated in Fig. 4. For the period 9 June 2020 – 16 May 2021, a leak near the inlet to the instrument was identified at the Hobbs site. The field calibration tank emptied unusually quickly, and flow rates measured

near the instrument inlet on a site visit indicated a leak. Although the leak is not apparent upon inspection of the data in isolation or via comparison with the other network sites, we have flagged this data as unsuitable for use and replaced with a placeholder value (NaN) in an abundance of caution. The original data is, however, available online.



**Figure 4: Data availability and instruments used at each site. Inlet level (meters above ground level) is indicated besides the site name. (a) Methane ($CH_4$) data availability at all sites. (b) Carbon dioxide ($CO_2$) data availability. (c) Hydrogen sulfide ($H_2S$) data availability. (d) Methane isotope ($\delta^{13}CH_4$) data availability. At the sites with more than one level of measurement (e.g., HOBB and CARL), all the levels operate using the same instrument at that site. Instruments replacements were made due to instrument failures at FORT, MALJ and NOTR.**

## 4.2 Diurnal cycle and seasonality

Hourly composites of the $CH_4$ and $CO_2$ mole fractions indicate clear seasonality (Fig. 5). The atmospheric boundary layer is typically deeper in the summer, which is consistent with the observed lower mole fractions of $CH_4$ and $CO_2$, compared to the wintertime. Methane and carbon dioxide have a distinct diurnal cycle during the summer months, with the highest mole fractions between 10 and 15 UTC (night). The observed $CH_4$ mole fraction in the summer is lowest at Fort Stockton, which is consistent with predominantly southerly winds (Fig. 1b) and the majority of the oil and natural gas facilities being to the northwest of this tower. During the winter months, when winds are predominantly from the SW – W direction, the lowest

CH$_4$ is measured at Maljamar on the northern edge of the network for most hours of the day. The diurnal amplitude in CO$_2$ is

1 ppm during the winter, compared to 5 ppm during the summer.



**Figure 5: Diurnal cycle of species measured at the Permian Basin network during the winter months (December, January, February), and during the summer months (June, July, August). (a) Diurnal cycle of CH$_4$ during winter. (b) Diurnal cycle of CH$_4$**
**during summer. (c) Diurnal cycle of CO$_2$ during winter. (d) Diurnal cycle of CO$_2$ during winter. All data are from the highest level available.**

Due to data availability, the seasonality of H$_2$S cannot be assessed. Hourly composites of H$_2$S mole fraction during the

summer months show a clear diurnal cycle (Fig. 6). The concentrations at 91 m AGL and inside the building have similar

Earth System
Science
Data

diurnal cycle signature and magnitude. On the other hand, 2 m AGL presented the highest concentration, and is significantly
different from the 91 m AGL observations. Filtration by the air conditioning system appears to have reduced the $H_2S$ mole
fractions inside the building. While background levels of $H_2S$ are essentially zero, the mean $H_2S$ mole fraction at direct
exposure level for humans (2 m AGL) is on the order of 8 ppb around 12 UTC, which is 4 times larger than the levels
observed inside the building and at 91 m AGL. These levels of $H_2S$ are not a health concern (OSHA, 2022), but do indicate
the existence of an $H_2S$ source near the tower.

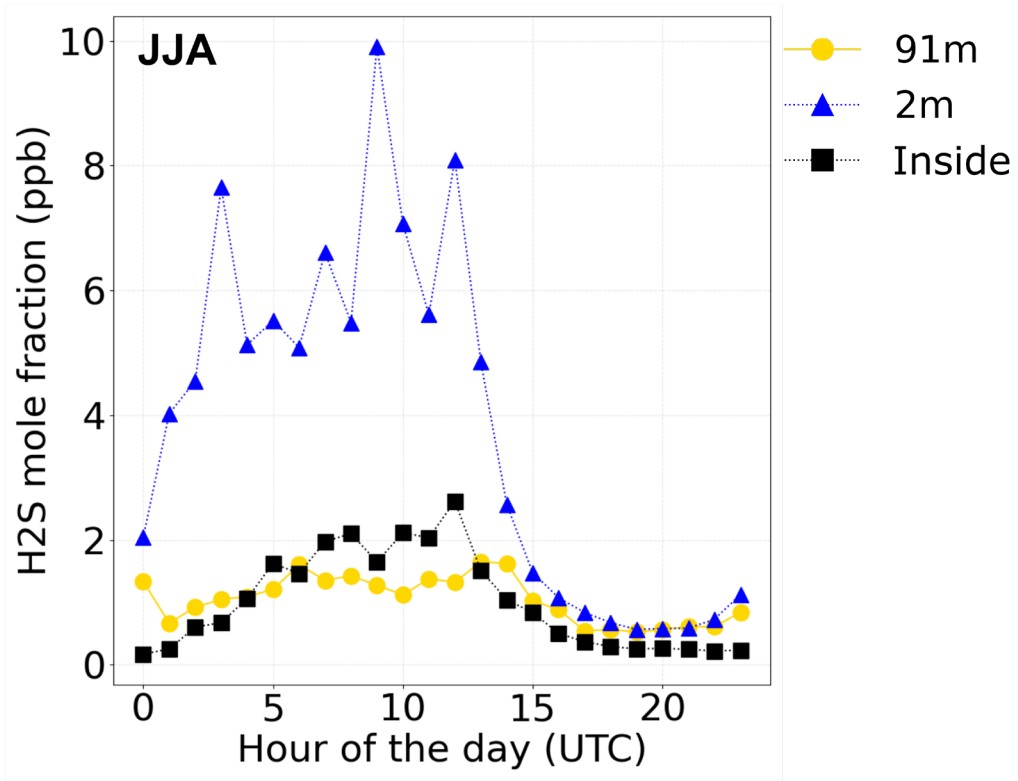

**Figure 6: Diurnal cycle of $H_2S$ measured at the Hobbs site during the summer months (June, July, August).**

**4.3 Methane, carbon dioxide, and hydrogen sulfide enhancements**

Atmospheric inversion techniques, used to estimate gas emissions, rely on accurate quantification of enhancements, which
are defined as the difference between the tower network background which may be defined in a variety of ways (e.g., Miles
et al., 2017; Karion et al., 2021; Sargent et al., 2018) and the mole fraction observed at each tower. Typically, enhancements
are calculated for afternoon hours, and here we defined afternoon hours from 20 to 23 UTC. To determine the enhancements,



we averaged the afternoon mole fractions at each tower, and obtained the background for $CH_4$ and $CO_2$ from the minimum averaged afternoon mole fraction of the entire network. Thus, each day has one value for the background, and each tower has one enhancement value per day. For $H_2S$, we assumed that the background is zero since this gas is not expected to be found naturally in this region. We used the wind data from Lea International Airport in Hobbs and excluded directions from which the wind originated for five or fewer days during the analysis period (e.g., summer and winter months). Calm winds (< 1.6

m/s) were also excluded from the analysis.

Enhancements of methane have strong seasonality, with smaller enhancements during the summer months (Fig. 7a) when compared to the enhancement during the winter months (Fig. 7b). During the summer, there is intense surface heating in the region, and deep boundary layer depths, compared to the winter months when more stable atmospheric conditions and lower

atmospheric boundary layer depths occur. The largest enhancements of $CH_4$ occur when the wind blows from the center of the Delaware sub-basin (Fig. 7), which is coincident with the high estimated emission rates of methane (Zhang et al., 2020).

**Figure 7: Methane enhancements at each tower location. The background for the figures indicates individual well locations for the**
**Permian Basin. (a) CH₄ enhancements during summer months, i.e., June, July, August (JJA). (b) CH₄ enhancements during**
**winter months, i.e., December January, February (DJF). The "triangles" represent the mean of the afternoon (20 - 23 UTC)**
**enhancements coming from the indicated direction. The gray boundary delimits the Permian Basin, while the black line is the**
**boundary between New Mexico and Texas. Credits for basemap: Esri, DigitalGlobe, GeoEye, Earthstar Geographics,**
**CNES/Airbus DS, USDA, USGS, AeroGRID, IGN, and the GIS User Community.**


The seasonality of methane observations is also apparent in the daily afternoon differences between the largest and smallest

CH₄ mole fraction measured from the tower network from 1 March 2020 to 1 August 2021 (Fig. 8). While in summer

months the afternoon differences do not exceed 150 ppb, in the winter months the differences reached more than 900 ppb. For the 30-day mean, the summer differences ranged from 50 to 100 ppb, and winter differences were twice the summer

values, and are usually between 150 and 200 ppb.

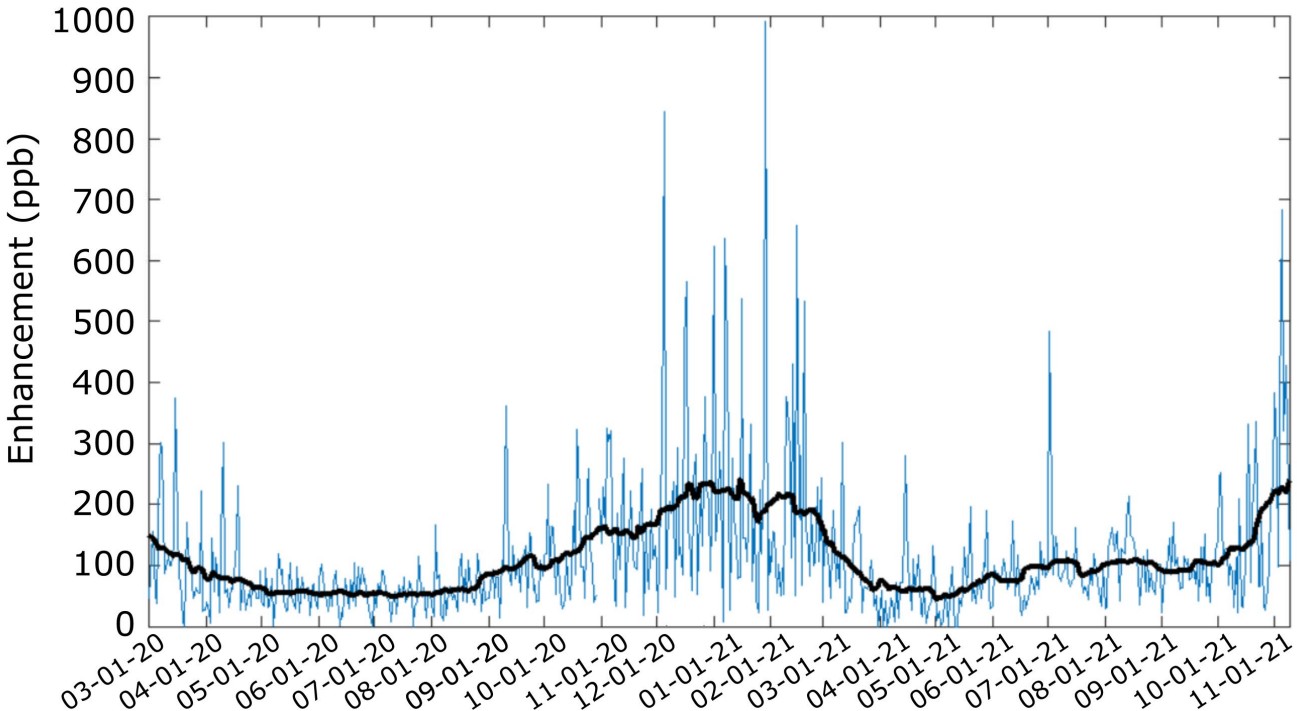

**Figure 8: Daily afternoon differences (blue line) between the largest and smallest CH₄ mole fraction measured from the tower network from 1 March 2020 through 9 November 2021. Afternoon values are calculated by averaging measurements between 20-**
**23 UTC. The black line indicates the 30-day running mean.**

Even though the composite diurnal cycle of $CO_2$ mole fractions presented some seasonality (Fig. 5c and 5d), the magnitude of $CO_2$ enhancements did not (Fig. 9). There were, however, changes in enhancements related to the seasonality of the wind direction (Fig. 9). We did not expect to observe significant enhancements of $CO_2$ coming from the O&G basin, but some

interesting patterns emerged. The observations revealed that at Notrees tower the enhancements were larger than 3 ppm with winds coming predominantly from South (Fig. 9b) and South-Southwest directions (Fig. 9a). The Notrees enhancements also corroborate the enhancements observed at Fort Stockton, which has the largest enhancements to SSW during the summer; and during the winter, Fort Stockton has enhancements coming from the North, pointing to a possible source that is between Notrees and Fort Stockton. As for Carlsbad and Maljamar, during the summer, the enhancements are more isotropic, from

NE to SSW, and the largest enhancements from both towers come from the directions of the city of Carlsbad. During the winter, Carlsbad, particularly, has a strong enhancement coming from the east.

**Figure 9: Carbon dioxide enhancements at each tower location. The background for the figures is the land cover from NLCD**
**(National Landcover database; MRLC, 2019) for the Permian Basin. (a) CO₂ enhancements during summer months, i.e., June,**
**July, August (JJA). (b) CO₂ enhancements during winter months, i.e., December January, February (DJF). The "triangles"**
**represent the mean of the afternoon (20 - 23 UTC) enhancements coming from the indicated direction. The gray boundary delimits**
**the Permian Basin, while the black line is the boundary between New Mexico and Texas. Credits for basemap: Esri, DigitalGlobe,**
**GeoEye, Earthstar Geographics, CNES/Airbus DS, USDA, USGS, AeroGRID, IGN, and the GIS User Community.**


Only one of the network towers has measurements of H₂S during the summer, as stated above, and thus we cannot verify the

seasonality of this dataset. However, the H₂S enhancements obtained from the Hobbs tower at 91 m AGL, in the order of 2

ppb, indicate a potential source NE of the tower (Fig. 10). There is a patch of oil and gas wells 0.5 – 1.5 km to the east and northeast of the tower that may be the source of H2S. The enhancements computed from the 2 m AGL inlet, not shown here,

presented the same pattern as the 91 m AGL, and are, on average, 0.05 ppb larger than the enhancements at the top level. During the summer, the enhancements obtained from the inlet temporarily installed inside the building, on average, did not exceed 0.2 ppb, coming from the southeast.

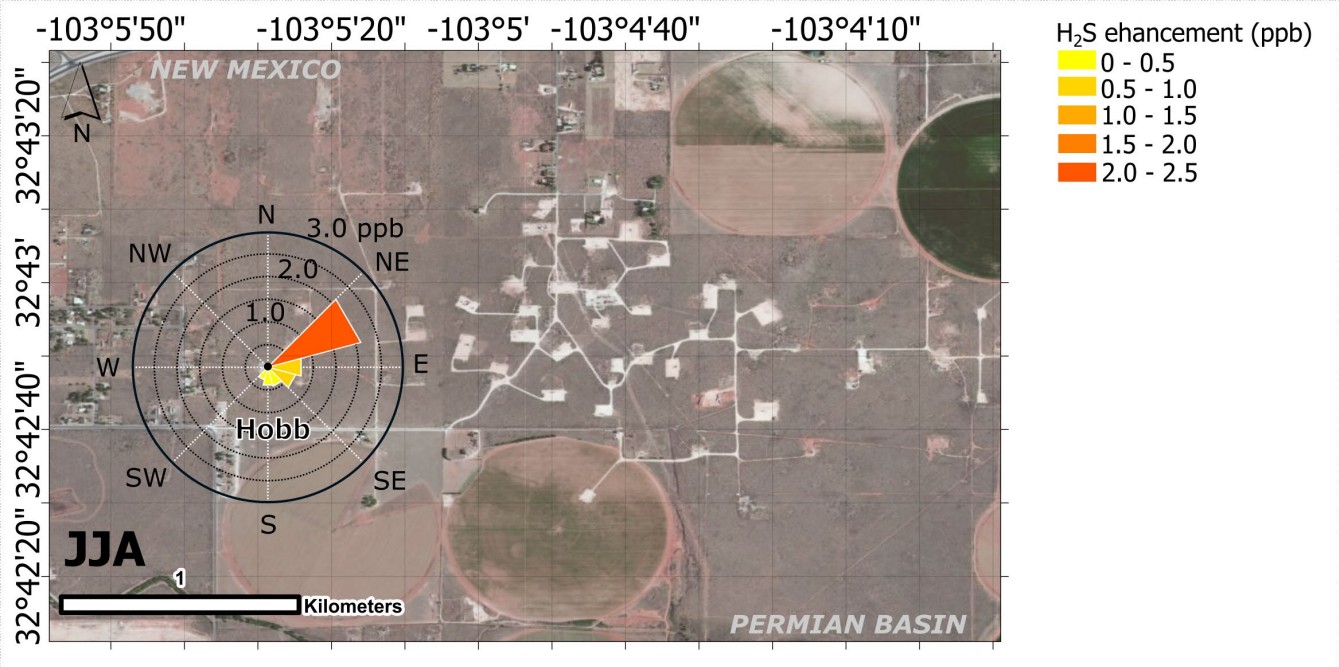

**Figure 10: Hydrogen sulfide enhancements at the Hobbs tower, at 91 m AGL, during summer months, i.e., June, July, August (JJA). The "triangles" represent the mean of the afternoon (20 - 23 UTC) enhancements coming from the indicated direction. Credits for basemap: Esri, DigitalGlobe, GeoEye, Earthstar Geographics, CNES/Airbus DS, USDA, USGS, AeroGRID, IGN, and the GIS User Community.**

**4.4 Isotopic ratio source signature**

We used the Keeling plot approach (Keeling, 1961; Röckmann et al., 2016; Miles et al., 2018), determining the intercept of the best-fit line of the isotopic ratio as a function of the inverse methane mole fraction, to estimate the isotopic ratio of the methane source at the Maljamar tower. The intercepts of the best-fit lines for the peaks (Fig. 11) indicate that the sources contributing to the peaks have a mean isotopic ratio of −40.8 ± 0.5 ‰. Oil and natural gas extraction is the only significant

source of local methane in this region (Maasakkers et al., 2016). The methane is lighter than that observed in the Marcellus region (−31.2 ‰; Miles et al., 2018), and similar to that observed in the Barnett region (−41.8 ‰; Milkov et al., 2020). The correlation coefficients were lower than that observed via a similar tower-based method in the Marcellus (Miles et al., 2018).



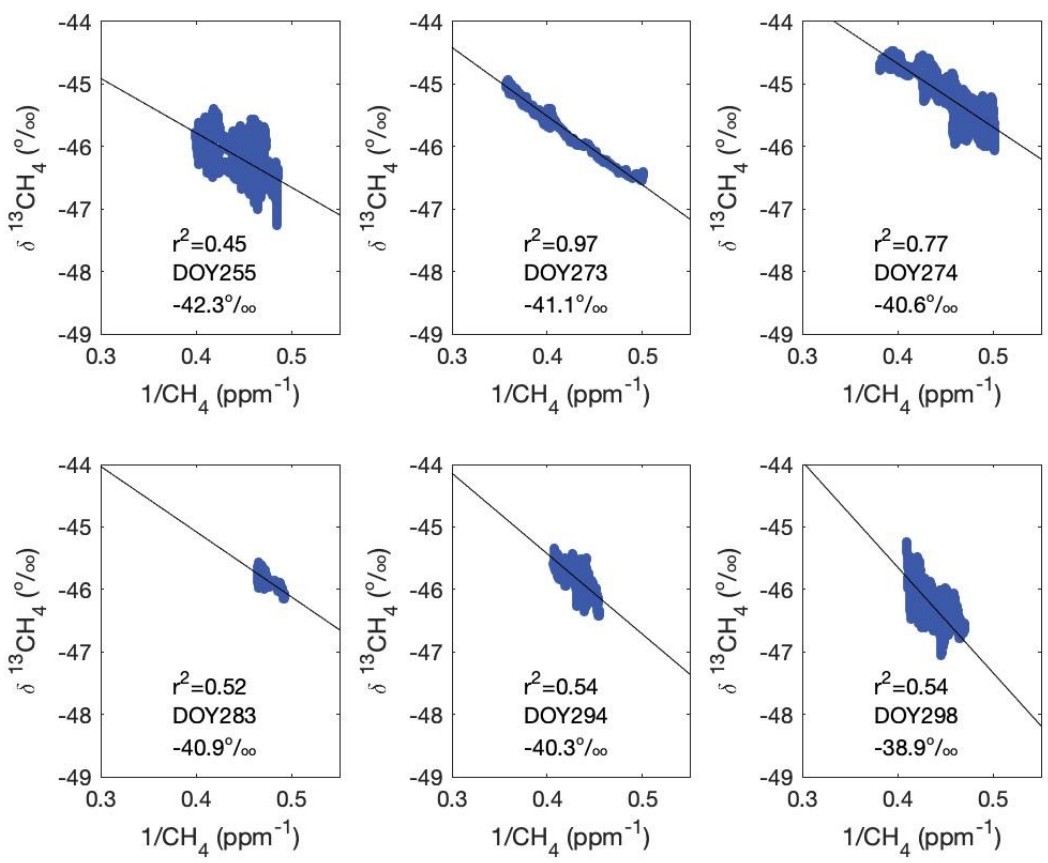


**Figure 11: Keeling plots for six CH₄ peaks measured at the Maljamar tower, using 10-min averaged data. Black lines indicate the best-fit lines. Correlation coefficients (r²), day of year (DOY), and y intercepts are indicated in the plots.**

## 5 Data availability

The data is available at The Pennsylvania State University Data Commons under DOI https://doi.org/10.26208/98y5-t941

(Monteiro et al., 2021). We plan to update the data repository annually.

The dataset is organized by yearly data files and named by the host institution identification (PSU), the project identification (PERMIAN), type of measurements (INSITU), tower identification (e.g., CARLSBAD, HOBBS, FORTSTOCKTON, MALJAMAR, NOTREES), and year. The earliest files start on 01 March 2021.


In the datasets, the columns include location code, instrument serial number, inlet height (m AGL), minimum time included in the hourly average (decimal day of year, UTC), maximum time included in the hourly average (decimal day of year, UTC), year, day of year, hour (UTC), calibrated $CO_2$ (dry mole fraction, ppm), standard deviation of the raw (2-3 s) $CO_2$ data within the hour (ppm), estimated $CO_2$ uncertainty for that hour (ppm), calibrated $CH_4$ (dry mole fraction, ppb), standard

deviation of the raw (2-3 s) $CH_4$ data within the hour (ppb), estimated $CH_4$ uncertainty for that hour (ppb), $H_2S$ (ppb) or $\delta^{13}CH_4$ (‰) (depending on instrument type), standard deviation of the raw (2 – 3 s) $H_2S$ or $\delta^{13}CH_4$ data within the hour, estimated uncertainty for that hour, and a user flag (1 = good, 0 = not recommended for use or not available).

Another sub-product of this dataset is the Permian Map website, developed by EDF (PermianMAP, 2021), providing access

to intermediate data products and a map of Permian Basin emissions, updated periodically.

## 6 Conclusions

The data presented show that regional tower networks can be operated to monitor methane emissions from O&G basins. Data quality and continuity successfully document regional methane enhancements associated with O&G production in the Delaware sub-basin of the Permian basin. The location of upwind/downwind sites both change significantly as a function of

season, illustrating the need to surround a basin with measurements. The magnitude of the enhancements also changes significantly vs. season, illustrating that accurate descriptions of boundary layer depths and winds are needed to interpret the data. A greater density of sites, more readily available instrument spares or more reliable GHG measurement instruments could increase the data density, but the existing network performed sufficiently to document the basic characteristics of enhancements associated with this production basin. Basins with more complex methane background conditions and/or

smaller emission rates may prove more challenging to characterize.

## Author contributions

SR and NM collected data. BJH performed data ingest and initial screening. NM performed quality control, data processing, and uncertainty analysis. VM performed data analysis and produced the majority of the figures. VM and NM wrote the manuscript, with contributions from KD, SR, and ZB. NM, KD, SR, ZB, and DL designed the study.

**Competing interests**

The authors declare that they have no conflict of interest.



**Acknowledgements**

The tower network and modeling work was completed as part of the PermianMAP project, supported by the Environmental Defense Fund and its donors, including Bloomberg Philanthropies, Grantham Foundation for the Protection of the
Environment, High Tide Foundation, the John D. and Catherine T. MacArthur Foundation, Quadrivium, and the Zegar Family Foundation. Computations for this research were performed on The Pennsylvania State University's Institute for Computational and Data Sciences' Roar supercomputer. We thank Carlsbad Caverns National Park for hosting an instrument.

**Financial support**

The PermianMAP project has been supported by the Environmental Defense Fund (award #224260) and its donors.

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
