# Peer review of "Methane, carbon dioxide, hydrogen sulfide, and isotopic ratios of methane observations from the Permian Basin tower network"

_Earth System Science Data, 2022_

## Referee Comment (RC1)

The work by Monteiro et al. presents data collected from the Permian Basin tower network, focused on $CH_4$, but also for $CO_2$, $H_2S$, and $CH_4$ isotopes, based on *in situ* operations of high-precision CRDS instrumentation. The Permian Basin is most certainly a region of high interest in terms of understanding and quantifying $CH_4$ emissions from natural gas production, and the data presented here is a significant contribution to quantifying emissions in this region. I find that the manuscript in its current form is mostly clear and well-written in describing the technical aspects of the dataset, and only have relatively minor, technical corrections to suggest prior to publication.

One important issue I would point out is that the data filed ("monteiro-et-al-permian-basin-in-situ-tower-greenhouse-gas-data-2021.zip) do not seem to be accessible when I've tried downloading it from the link provided by the authors (the message I see is "Server Error", "401 - Unauthorized: Access is denied due to invalid credentials. You do not have permission to view this directory or page using the credentials that you supplied"). I trust that this error will be corrected by the authors.

Line 31. "2" in $CO_2$ should be subscripted.

LIne 40. Another important "relatively diffuse, large-area, low-intensity" source of $CH_4$ worth mentioning may be leaks in residential/commercial natural gas consumption post metering (i.e. In pipes inside homes and buildings). See Wennberg et al. 2012 (10.1021/es301138y).

Figure 1. Could a zoomed-out map of the region be provided to orient the basin to the general region (e.g. A map large enough to show more of Texas)? Also, I was wondering about known well locations, which are shown later plot 7, perhaps add a line in the caption to suggest that well locations are shown in later plots?

Line 148. Note that, as far as I can tell, the sampling sequence for the two inlet heights (and lab observations, in the case of $H_2S$?) have not been specified anywhere in the text. This information should be added somewhere, perhaps in section 3.1 (which might require a slight adjustment in the section title)?

Line 150. Can the authors clarify why different lengths of the Nafion dryers were used? I'm assuming this was to account for the different flow rates of the Picarro instruments?

Line 153. The regulator part# specified here (51-14 A-590) currently does not seem to adequately identify the regulator, as nothing under this part# comes up in my search. My guess is that the regulator referred to here is the Model 14 (https://industry.airliquide.us/sites/activity_us/files/2015/10/08/nickel_plated_pressure_regulators_14.pdf), or perhaps the Model 14A, a low-flow version of the Model 14 (https://industry.airliquide.us/sites/activity_us/files/2015/10/09/low_flow_regulator_14a.pdf). Somewhat confusingly, Airgas (now another subsidiary of Air Liquide) apparently uses the part# Y12114C590-AL for the Model 14, although I've not seen this part# listed directly on an Airgas catalog/website. If my guess is correct, my suggestion is that you refer to them as Model 14 (or 14A?) regulators, and use the Air Liquide part# for them, where a web search turns up something useful.

Line 156. Just to clarify, the $H_2O$ correction referred to are those reported by the Picarro instrument, based on the factory default $H_2O$ correction factors, as opposed to those based on instrument-specific $H_2O$ correction factors and/or some $H_2O$ correction applied in post-processing, correct? The wording here seems a little vague, and could be clarified.

Line 163. Regarding the field calibration tanks, can the authors specify the size of the tanks, and their expected lifetime in the field? Also, how were the field tanks calibrated?

Figure 2. I will note that the sampling for multiple inlet heights are not shown in this figure. I've found that similar figures in Richardson et al. 2017 do show how the multiple inlets are configured, and suggest revising Figure 2 in a similar way.

LIne 174. The residence times listed here seem long, especially for the G2132-i where the data interpretation may be significantly affected by a 45-70min delay. If there was a good reason for not having an additional air pump to reduce the residence times in the air lines, a common practice when long intake lines are used, then I feel the authors should state this reason. Otherwise, I think the authors should at least state that the delay due to the long residence time should be considered if performing time-matched analysis with other measurements and/or models.

Figures 5, 6: Please add subscripts to $CO_2$, $CH_4$, and $H_2S$.

Line 300. Background determination is certainly a very subjective and difficult issue, and I do feel that a lengthy scientific discussion into the merits of the method presented by the authors is probably beyond the scope of this work, which focuses on presenting the data themselves.

However, since the authors discuss this baseline here, I do hope that the baseline values calculated with this method are made available in the dataset (again, my access was limited so I couldn't check this directly).

Figure 5 vs Figures 7,8. I'm having a somewhat difficult time reconciling the diurnal variability presented for $CH_4$ in Figure 5 vs the $CH_4$ pollution magnitudes shown in Figures 7 and 8. Just comparing figures 5(a) and 5(b), I would have thought that the enhancement magnitudes would be larger in summer since the background values (mostly from FORT?) are significantly lower in the summer months. Am I to assume that the large afternoon enhancements seen during the wintertime in Figure 8 are more sporadic in nature and not widespread/frequent enough to be a factor in Figure 5, or is there processing in Figure 5 to remove large outliers that the authors have not explained? Or perhaps, there's large standard deviations in the hourly means in Figure 5 that are not apparent when only looking at the mean? Perhaps some sentences that put the enhancements shown in Figure 8 in context with the overall observed trends would be helpful.

Line 354. Subscript missing in $H_2S$.

---

## Author Response (AR1)

**REFEREE #1**

The work by Monteiro et al. presents data collected from the Permian Basin tower network, focused on $CH_4$, but also for $CO_2$, $H_2S$, and $CH_4$ isotopes, based on in situ operations of high-precision CRDS instrumentation. The Permian Basin is most certainly a region of high interest in terms of understanding and quantifying $CH_4$ emissions from natural gas production, and the data presented here is a significant contribution to quantifying emissions in this region. I find that the manuscript in its current form is mostly clear and well-written in describing the technical aspects of the dataset, and only have relatively minor, technical corrections to suggest prior to publication.

Thank you for your kind words and for your time in completing your review.

One important issue I would point out is that the data files ("monteiro-et-al-permian-basin-insitu-tower-greenhouse-gas-data-2021.zip) do not seem to be accessible when I've tried downloading it from the link provided by the authors (the message I see is "Server Error", "401 - Unauthorized: Access is denied due to invalid credentials. You do not have permission to view this directory or page using the credentials that you supplied"). I trust that this error will be corrected by the authors.

There was a recent temporary outage of the entire Penn State Data Commons system. Apologies.  It appears to be working now.  We tested on multiple computers and multiple browsers.

Line 31. "2" in $CO_2$ should be subscripted.

We corrected this typo and also searched the remainder of the document for any additional cases of unsubscripted $CO_2$, $CH_4$, and $H_2S$.

Line 40. Another important "relatively diffuse, large-area, low-intensity" source of $CH_4$ worth mentioning may be leaks in residential/commercial natural gas consumption post metering (i.e. In pipes inside homes and buildings). See Wennberg et al. 2012 (10.1021/es301138y).

We added this reference.

Figure 1. Could a zoomed-out map of the region be provided to orient the basin to the general region (e.g. A map large enough to show more of Texas)?

Thank you for the suggestion. We added a zoomed-out map to complement Figure 1.

Also, I was wondering about known well locations, which are shown later plot 7, perhaps add a line in the caption to suggest that well locations are shown in later plots?

We added an overview map, as suggested.  We also added a sentence in the caption, "Known well locations are shown in Fig. 7."

Line 148. Note that, as far as I can tell, the sampling sequence for the two inlet heights (and lab observations, in the case of $H_2S$?) have not been specified anywhere in the text. This information should be added somewhere, perhaps in section 3.1 (which might require a slight adjustment in the section title)?

Thank you for pointing out this omission.  We added the information concerning the sampling sequence and updated the section title.

Line 150. Can the authors clarify why different lengths of the Nafion dryers were used? I'm assuming this was to account for the different flow rates of the Picarro instruments?

Differing lengths of Nafion dryers were used based simply on availability.  We added a phrase to the text.

Line 153. The regulator part# specified here (51-14 A-590) currently does not seem to adequately identify the regulator, as nothing under this part# comes up in my search. My guess is that the regulator referred to here is the Model 14 (https://industry.airliquide.us/sites/activity_us/files/2015/10/08/nickel_plated_pressure_regulators_14.pdf), or perhaps the Model 14A, a low-flow version of the Model 14 (https://industry.airliquide.us/sites/activity_us/files/2015/10/09/low_flow_regulator_14a.pdf). Somewhat confusingly, Airgas (now another subsidiary of Air Liquide) apparently uses the part# Y12114C590-AL for the Model 14, although I've not seen this part# listed directly on an Airgas catalog/website. If my guess is correct, my suggestion is that you refer to them as Model 14 (or 14A?) regulators, and use the Air Liquide part# for them, where a web search turns up something useful.

We clarified this to read, "Scott Specialty Gas (now Air Liquide) two-stage regulators (part number 51-14 A-590, similar to Air Liquide, part number Q1-14B-590) were used for sampling the field calibration tanks."

Line 156. Just to clarify, the $H_2O$ correction referred to are those reported by the Picarro instrument, based on the factory default $H_2O$ correction factors, as opposed to those based on instrument-specific $H_2O$ correction factors and/or some $H_2O$ correction applied in postprocessing, correct? The wording here seems a little vague, and could be clarified.

Yes, only the standard $H_2O$ corrections reported by the Picarro instrument were completed. We added a clarifying phrase.

Line 163. Regarding the field calibration tanks, can the authors specify the size of the tanks, and their expected lifetime in the field? Also, how were the field tanks calibrated?

We used N80 tanks, which with our sampling strategy, should last 12 - 18 months. To calibrate the field tanks prior to deployment, we first calibrated a laboratory Picarro using four NOAA tertiary tanks, sampling for at least 5 min after equilibration, and repeated the sampling at least once. We then sampled each field calibration tank similarly and assigned the average to each tank. We added these details to the text. We also clarified that while we calibrated the standard $CH_4/CO_2/H_2O$ instruments prior to deployment, for the $CH_4/H_2S/H_2O$ and $\delta^{13}CH_4$ instruments, we characterized them and applied the calibration in post-processing.

Figure 2. I will note that the sampling for multiple inlet heights are not shown in this figure. I've found that similar figures in Richardson et al. 2017 do show how the multiple inlets are configured, and suggest revising Figure 2 in a similar way.

We only used multiple heights at one of the locations in the network, so we left the figure as is. We added a clarifying sentence to the caption to explain that the Hobbs instrument is set up to sample at multiple heights as in Richardson et al., 2017.

LIne 174. The residence times listed here seem long, especially for the G2132-i where the data interpretation may be significantly affected by a 45-70min delay. If there was a good reason for not having an additional air pump to reduce the residence times in the air lines, a common practice when long intake lines are used, then I feel the authors should state this reason. Otherwise, I think the authors should at least state that the delay due to the long residence time should be considered if performing time-matched analysis with other measurements and/ or models.

This is an excellent point.  We should add additional pumps to reduce the lag times, particularly for the (low flow) isotopic instruments.  We added a sentence describing that we have added the approximate delay time to the data, such that the time stamp represents the sampling time, not the analysis time.

Figures 5, 6: Please add subscripts to $CO_2$, $CH_4$, and $H_2S$.

We corrected the figures' subscripts and also searched the remainder of the document for any additional cases of unsubscripted $CO_2$, $CH_4$, and $H_2S$.

Line 300. Background determination is certainly a very subjective and difficult issue, and I do feel that a lengthy scientific discussion into the merits of the method presented by the authors is probably beyond the scope of this work, which focuses on presenting the data themselves. However, since the authors discuss this baseline here, I do hope that the baseline values calculated with this method are made available in the dataset (again, my access was limited so I couldn't check this directly).

We agree - there are many ways to calculate background and it is a topic of debate in the literature.  The method presented here is very simple, and easy to calculate - the minimum afternoon value for the network, regardless of wind direction - so we did not include it in the dataset.

Figure 5 vs Figures 7,8. I'm having a somewhat difficult time reconciling the diurnal variability presented for $CH_4$ in Figure 5 vs the $CH_4$ pollution magnitudes shown in Figures 7 and 8. Just comparing figures 5(a) and 5(b), I would have thought that the enhancement magnitudes would be larger in summer since the background values (mostly from FORT?) are significantly lower in the summer months. Am I to assume that the large afternoon enhancements seen during the wintertime in Figure 8 are more sporadic in nature and not widespread/frequent enough to be a factor in Figure 5, or is there processing in Figure 5 to remove large outliers that the authors have not explained? Or perhaps, there's large standard deviations in the hourly means in Figure 5 that are not apparent when only looking at the mean? Perhaps some sentences that put the enhancements shown in Figure 8 in context with the overall observed trends would be helpful.

Thank you for pointing out this difference between summer and winter.  As you alluded to, in the summer, FORT is the site with minimum afternoon CH4 50% of the days, with the other sites having the minimum at most 17% of the days.  During the winter, the site exhibiting the minimum $CH_4$ is much less consistent, with CARL at 29%, FORT at 27%, and MALJ at 20% of the days.  Thus while the intra-network differences are large in the winter, averaging about 200

ppb (Fig. 8), the mean difference between sites is 40 ppb (Fig. 5b).  We added this context to the description of Fig. 8 to clarify:  "Note that Figure 5 is averaging across a 3-month period while Figures 7 and 8 are daily maximum and minimum differences. In the summer, FORT is the site with minimum afternoon $CH_4$ 50% of the days, but during the winter, the site exhibiting the minimum $CH_4$ is much less consistent, with CARL at 29%, FORT at 27%, and MALJ at 20% of the days.  Thus, while the daily afternoon intra-network differences are large in the winter (Fig. 8), the mean difference between sites when averaged over 3 months is 40 ppb (Fig. 5b)."

Line 354. Subscript missing in $H_2S$.

We corrected the figures' subscripts and also searched the remainder of the document for any additional cases of unsubscripted $CO_2$, $CH_4$, and $H_2S$.

**REFEREE #2**

General comments:

Monteiro et al. present the core elements of a new in-situ network, which was established to monitor atmospheric $CH_4$, $CO_2$, $H_2S$ and $\delta^{13}CH_4$ in the Permian Basin. They detail: station design, instruments used, measurement and calibration procedures as well as data processing and flagging rules in detail. The dataset is made publicly available and the manuscript also includes a first order analysis of results, discussing relevant quantities such as diurnal cycles for each seasonn, intra-network mixing ratio gradient changes over time as well as the relationship between site level $CH_4$ mixing ratios and meteorological conditions. Overall, the manuscript is well-written and sections are clearly structured.  This manuscript fits well within the scope of ESSD and should be considered for publication after some minor and technical comments have been addressed.

We appreciate your comments and time to complete the review.

Specific and technical comments:

L14: change to 'species'

We changed to 'species'.

L17: suggest to change to: 'prior *to* deployment' or 'before deployment'

We changed to 'prior to deployment'.

L24: Which tower are you referring to here? There are multiple towers in this network.

There is only one tower with $H_2S$ measurements (Hobbs, New Mexico). We added the specific location in the text for clarification.

L31: change to $CO_2$

We corrected the subscript.

L34: please consider clarifying this statement/ How much reduction would be considered 'dramatic' 50%, 80%? What does 'near-term' mean here? 1 year, 10 years, 50 years?

We clarified this statement in the text adding the following sentence: "According to Ocko et al (2021), full implementation of all methane abatement technologies that are already technically feasible could cut anticipated global methane emissions in 2030 by 57%, and global-mean average methane warming rates between 2030 and 2050 would consequently be reduced by 26%."

L47: consider clarifying what 'these basins' refers to here

We substitute the sentence for: "This expansion of O&G production has prompted an increasing interest in monitoring of methane emissions from O&G basins on regional scales for the purpose of possible regulations and commercial incentives for operators to prove low emissions."

L72: This is a good summary of different O&G monitoring techniques. However, you only mention Lin et al. 2021 for in-situ monitoring, while Chan et al. 2020, preceded this work and demonstrated the ability of in-situ monitoring to be used to quantify $CH_4$ emissions for the Western Canadian Sedimentary Basin spanning Alberta and Saskatchewan:

Thank you for pointing out this omission. We edited the sentence to read, "Chan et al. (2020) used in-situ monitoring to quantify methane emissions for the Western Canadian Sedimentary O&G Basin and Lin et al. (2021) quantified methane emissions from the Uinta O&G Basin."

L110, figure 1: please add a symbol indication the location of Lea County Regional Airport or give an indication of its distance from the HOBB site. Please also update the labels of the 5 tower sites to match the manuscript (all uppercase letters). Furthermore, please consider adding information about the facility (e.g. O&G wells) locations or other $CH_4$ emission priors to sub-figure (a) or at least refer to figure 7 here. It is very hard to judge if the network locations is suitable without knowing where emissions are to be expected.

We added the airport location and updated the tower's labels (uppercase). Changes can be seen in the updated Figure 1b. We referred to Figure 7 to indicate known O&G wells location (added in figure caption).

L125, table 1: please add the data of the move/re-install of CARL station in the Install data column.

We added the move/re-install date (Table 1).

L189: Presumable the $\delta^{13}CH_4$ data is reported on the VPDB scale or some equivalent?

The isotopic ratios were tied to the Vienna Pee Dee Belemnite (VPDB) scale. We added a phrase to this effect.

L268: Please elaborate, what are 'hourly composites'? Are you referring to the average, median, mode of the measurement distribution gathered within one hour?

We changed this phrase to clarify. The text now says, "Composited means of hourly $CH_4$ and $CO_2$ mole fractions (averaged over summer and winter seasons)...".

L299: Consider adding the information on local time (especially for non-US readers), i.e. what is 20 – 23 UTC in LT?

We added a sentence clarifying the conversion from UTC to local time within the text (Line 297 from the reviewed manuscript): "In Texas, the local time is UTC-5 and in New Mexico, UTC-6.".